# Impact of Precision Feeding During Gestation on the Performance of Sows over Three Cycles [note 1]

**DOI:** 10.3390/ani14233513

**Published:** 2024-12-05

**Authors:** Laetitia Cloutier, Lucie Galiot, Béatrice Sauvé, Carole Pierre, Frédéric Guay, Gabrielle Dumas, Patrick Gagnon, Marie-Pierre Létourneau Montminy

**Affiliations:** 1Centre de Développement du Porc du Québec Inc., 815 Rte Marie-Victorin, Lévis, QC G7A 3S6, Canada; lcloutier@cdpq.ca (L.C.); lgaliot@cdpq.ca (L.G.); bsauve@cdpq.ca (B.S.); gdumas@cdpq.ca (G.D.); pgagnon@cdpq.ca (P.G.); 2Département des Sciences Animales, Université Laval, 2425 rue de l’Agriculture, Québec, QC G1V 0A6, Canada; carole.pierre.1@ulaval.ca (C.P.); frederic.guay@fsaa.ulaval.ca (F.G.)

**Keywords:** precision feeding, sow, lysine, gestation, bump feeding, longevity, post-weaning, nitrogen, phosphorus

## Abstract

Throughout gestation, sows are conventionally fed one diet formulated to meet the requirements of most sows. However, those diets can lead to nutritional excesses or deficiencies based on each sow’s parity or stage of gestation. To overcome this, one strategy currently employed is bump feeding, which involves increasing feed intake for sows at the end of gestation to address their higher nutritional requirements during this period. Another approach is precision feeding, which utilizes two types of feed simultaneously—one rich and the other low in nutrients—mixed in varying proportions according to the sow’s parity and stage of gestation. The objective of this project was to validate the impact of both bump feeding and precision feeding in comparison to conventional feeding on the reproductive performance of sows monitored over three cycles of gestation and lactation. Results indicated that bump feeding did not significantly enhance sow performance during lactation. However, precision feeding was found to reduce nitrogen intake by 10–13% and total phosphorus intake by 6–9% while also decreasing piglet mortality during lactation. These findings suggest that precision feeding can be a sustainable strategy for gestating sows, improving lactation performance, and reducing the environmental impact of swine production.

## 1. Introduction

In commercial farms, sows are generally fed a constant-nutrition feed throughout gestation. However, the nutrient requirements of sows vary with their stage of gestation and individual characteristics, such as age, parity, weight, or body condition [1,2]. The mismatch between feed intake and the requirements of gestating sows can cause significant fluctuations in body reserves, which can affect reproductive performance and longevity [3,4]. To address this issue, the “bump feeding” strategy is often applied, where the feed amount is increased towards the end of gestation to meet the higher nutrient requirements during that stage. However, the impacts and benefits of this strategy vary between studies [5]. The bump feeding strategy can be applied in two ways: (1) giving extra feed during the end of gestation, which increases energy intake compared to a diet without bump feeding, or (2) redistributing the feed intake during gestation without giving more energy than a diet without bump feeding. The first bump feeding strategy increases feeding costs [6,7,8], has few benefits [8], and results in excessive body conditions, which negatively impact the farrowing process in multiparous sows [6,9] and the feed intake of gilts during lactation [6,7,10]. However, some studies have observed that, for gilts, piglet birth weight increases with this strategy [11]. Few studies have been conducted on the second method of bump feeding, where there is no overconsumption of energy. Our current study focuses on the second method because it does not involve additional feeding costs, and positive effects have been observed on the farrowing process and piglet vitality [9]. We expect that bump feeding without overconsumption of energy will also avoid the negative effects related to excess weight gain in sows and gilts.

Precision feeding is another interesting strategy that uses two feeds, one low and one high in nutrients, mixed daily in different proportions to match the nutritional intake to the ongoing requirements of each animal. Studies in growing pigs [12] have indicated that this approach reduces feeding costs and simultaneously decreases environmental impact. Recent studies of precision feeding during gestation in sows have addressed the growing interest in this strategy, considering the changes in sow management from individual to group housing and advanced knowledge about precision feeding [13,14,15]. The development of sow feeding systems that can apply this feeding strategy has also contributed to recent advances [16,17]. The main advantage of precision feeding for gestating sows is the reduction in excess nutrient intake and, consequently, the excretion of nutrients such as nitrogen and phosphorus; conventional feeding often leads to nutrient excess for most sows [18,19,20]. A previous study indicated potential advantages of precision feeding for sows, initially gilts, monitored over two cycles of gestation and lactation, showing a reduced proportion of stillborn piglets and increased litter weight gain [21].

Here, we hypothesize that precision feeding during gestation will improve sow performance, particularly when starting at the first gestation of a sow. The objectives of this study are to validate the impact of a) bump feeding and b) precision feeding during gestation on the performance and body conditions of sows and on the post-weaning performance of piglets over three cycles.

## 2. Materials and Methods

### 2.1. Animal and Experimental Treatments

This study was carried out in accordance with the Canadian Council on Animal Care guidelines (CCAC; https://ccac.ca/en/certification/, accessed on 1 February 2021). The protocol was approved by the Animal Use and Care Protection Committee of Laval University (protocol number: 2021-867). The trial was conducted at the Research and Training Maternity Unit of Centre de Développement du Porc du Québec (CDPQ), located in Armagh, Québec, Canada. Sows from 4 batches (125 sows/batch), all gilts at start, were studied over 3 complete reproductive cycles from breeding to weaning, receiving the same treatment over those 3 cycles. The trial ran from February 2021 to July 2022.

Four isoenergetic treatments (2220 kcal of NE/kg feed) were compared: two conventional control treatments and two precision feeding treatments. The conventional treatments had a constant standardized ileal digestible lysine content (SID Lys) (0.53% SID Lys) throughout gestation: one treatment with a constant feed quantity throughout gestation (2.43 kg/d for gilts and 2.55 kg/d for multiparous sows; flat feeding, FF) and the other with a lower feed intake before 90 days of gestation (2.25 kg/d for gilts and 2.43 kg/d for multiparous sows) followed by higher feed intake thereafter (2.95 kg/d for gilts and 3.05 kg/d for multiparous sows; bump feeding, BF; average intake was identical to that of FF over the entire gestation). The two precision feeding strategies were based on the InraPorc model [18,22], considering either performance per parity (precision feeding per parity, PFP) or performance per parity with the individual weight of sows at breeding included in the model (precision feeding individual, PFI; Figure 1) to estimate requirements for SID Lys. Feed quantities (and thus energy) were determined using existing recommendations for the herd’s genetics (Large White × Landrace crossbreed, Alphagene, Saint-Hyacinthe, QC, Canada) [23]. The contents of amino acids other than Lys were adjusted based on the SID levels using the InraPorc model [22]. Phosphorus and calcium requirements were calculated according to Bikker and Block [24], and their content in feed was optimized to meet requirements while minimizing excess. The calculation of the SID Lys requirement for the PFP treatment was based on targeted performance objectives per parity; the same parameters were used for the PFI treatment except for breeding weight, where individual data were used instead of average weight per parity. For the PFP treatment during the time between breeding and transfer to group gestation housing on day 28 of gestation, the SID Lys content was fixed at 0.44% for all sows, simulating the use of a single feed during a period when precision feeding systems are generally not available in commercial farming. For PFI sows, precision feeding started immediately after breeding (Figure 1). During the first 28 days of gestation, for all treatments, adjustments to the allocated feed quantity were made based on weight and backfat thickness for gilts at breeding (140–149 kg and <12 mm, +300 g; 135–139 kg and 12–16.9 mm, +300 g; 135–139 kg and <12 mm, +600 g; <134 kg, +600 g) and on backfat thickness only for multiparous sow (12–15 mm, +300 g/day; <12 mm, +600 g).

### 2.2. Feeds

Two feeds (A and B) with different amino acid, calcium, and phosphorus contents were mixed during the experiment to create the feed for the four treatment groups (Table 1). The ingredient composition of the feeds remained fixed throughout the trial. Gestal 3G2 feeding stations (JYGA Technologies, St-Lambert-de-Lauzon, Québec, QC, Canada) automatically adjusted the proportions of each of the two feeds based on the gestation stage and the individual sow. A lactation feed corresponding to a conventional nutritional composition was automatically provided to all sows from farrowing to weaning with a Gestal Quattro feeding station. For this feed, the SID Lys content was 1.0% and the net energy was 2540 kcal/kg. Samples of each feed were collected weekly, and composite samples were analyzed each month to validate the nutritional composition in animo acids (Eurolysine, Paris, France) and total phosphorus and calcium (IRDA, Québec, QC, Canada).

### 2.3. Measurements

Live weight and backfat thickness were measured via ultrasound at the last rib (Ultra Scan 50 device, Alliance Medical Inc., Limerick, Ireland) by an accredited CDPQ technician 5 days before artificial insemination at entry into the farrowing room (110 days of gestation) and at weaning (21 days of lactation). Gains between these periods were calculated. Daily individual feed distribution was continuously recorded by the feeding system during gestation and lactation. Farrowing was not induced, and assistance was provided only when necessary. The average lactation duration was 21 days. The total number of piglets born (alive and stillborn) was counted, and individual piglet weights were measured at birth and upon death or adoption. Cross-fostering between sows of the same dietary treatment was allowed to equalize the litters and was performed after birth performance measurements within 24 h. Litter weights were measured at weaning. Litter weight gains were calculated, accounting for variations due to adoptions and mortalities. At days 21 of gestation, 112 of gestation, and 21 of lactation, blood samples from 6 sows per treatment for 3 batches were collected at the jugular vein to measure calcium and phosphorus status in the plasma using an ELISA kit (QuantiChrom™ Phosphate and Calcium Assay Kit, BioAssay Systems, Hayward, CA, USA).

For one batch per cycle, 5 litters from each of the 4 treatment groups were selected at weaning; within these litters, 5 representative piglets were chosen, totaling 100 piglets per group. The piglets were sent to the Deschambault Animal Science Research Center (CRSAD) site and housed in 20 pens (5 piglets per pen from the same litter). They were fed according to a commercial 3-phase feeding program (Agri-Marché, Québec, QC, Canada), with each phase lasting 14 days (Phase 1: 19.7% crude protein (CD), 1.30% Lys DIS, 0.51% P dig and 0.80% total Ca; Phase 2: 20.4% CD, 1.35% Lys DIS, 0.50% P dig and 0.75% total Ca; Phase 3: 19.8% CD, 1.27% Lys DIS, 0.41% P dig and 0.65% total Ca). All piglets were weighed at the beginning and end of the trial to evaluate average daily gain (ADG). Feed supply for each pen was weighed every day, and wastes were weighed at the end of each feeding phase to evaluate average daily feed intake (ADFI) per pen. One piglet per pen was scanned using dual-energy X-ray absorptiometry (DXA; Discovery W, Hologic, Marlborough, MA, USA) at the end of the first phase (day 14) and at the end of the third phase (day 42) to evaluate lean and fat deposition and bone mineral content (BMC).

### 2.4. Statistical Analysis

For each variable of interest regarding sow body condition and litter performance, mixed models were performed on gilts only (n = 502), on sows present in 2 consecutive cycles (n = 340) and on sows present in 3 consecutive cycles (n = 274), meaning that these sows never skipped batches. Analyses on nonproductive days were performed on all sows that completed three cycles, including those that skipped batches (n = 391). Analyses on longevity included all sows that completed at least the first cycle (n = 502).

Statistical models for the aforementioned variables were performed using the lmer and glmer functions from the lme4 package, and multiple comparisons were conducted using the emmeans package in R software version 4.2.2 (accessed on 1 July 2024) [26]. The fixed effect in the model was the treatment (FF, BF, PFP, or PFI), and the batch was considered a random effect. Repeated measurements for plasma analysis were used to test the fixed effect of the treatment, the parity, the time, and their interactions. For these analyses, the sow/litter was considered the experimental unit.

For the post-weaning analyses, the pen of 5 piglets from the same litter was considered the experimental unit for growth performance, and the block was the random effect (1 treatment/block). BW at weaning was introduced as a covariable in all the variables tested. For body composition, an individual piglet was considered the experimental unit. The statistical model included the fixed effect of treatment (FF, BF, PFP, or PFI) received by the sow, sow parity, and their interactions. All post-weaning statistical analyses were carried out using the mixed model (Minitab, version 21, State College, PA, USA, accessed on 1 July 2024).

Assumptions required for the statistical tests (residual normality and variance homogeneity) were checked in all cases, and transformations were applied when necessary. Model effects were considered significant for *p* ≤ 0.05 and trends for 0.05 < *p* ≤ 0.10.

## 3. Results

This trial took place during the settlement of the new CDPQ research barn. During the first gestation cycle, an episode of Mycoplasma hyosynoviae occurred, causing a higher occurrence of lameness in gilts, affecting all gilts of all treatments, and so they all received lincomycine (Lincomix^®^, Zoetis Canada Inc., Kirkland, QC, Canada), an antibiotic, in their feed at the same rate.

The quantities and proportions of feed A distributed during gestation are presented in Table 2, with the SID Lys, digestible phosphorus (P), and total calcium (Ca) intake of sows according to gestation stage and parity under different treatments. As expected, the FF and BF treatments received similar amounts of feed and nutrients throughout gestation. However, during the early stages of gestation, the SID Lys intake, digestible P intake, and total Ca intake were higher in the FF treatment than in the BF treatment. The BF treatment exhibited the opposite pattern, with lower intake before 90 days of gestation and a higher intake afterwards when compared with the FF treatment. In the precision feeding treatments, SID Lys intake for the PFP and PFI treatments, respectively, was 8% and 10% lower during the first cycle, 14% and 16% lower in the second cycle, and 18% and 23% lower in the third cycle when compared with the average intake for the FF and BF treatments. Throughout gestation, the SID Lys intake for precision feeding treatments was lower than for the FF and BF treatments during the first two stages but higher from days 90 and 110. From days 90 to 110 of gestation, SID Lys intake was higher for the PFP and PFI treatments by 25% during the first cycle, 20% during the second cycle, and 12% during the third cycle when compared with the BF and FF treatments.

The nutrient intakes for the PFP and PFI treatments were similar from days 29 to 110 of gestation across the three cycles. The main difference between the PFP and PFI treatments occurred during the first 28 days of gestation, with SID Lys intake, digestible P, and total Ca being the lowest in the PFI treatment, as expected.

### 3.1. Effects on Sow Performance

#### 3.1.1. Cycle 1

Live weight and backfat thickness of sows at breeding, after farrowing, and at weaning did not differ between treatments (Table 3). Sows in the PFP treatment group had a higher weight gain during gestation than in other treatments (*p* = 0.01), but no differences were observed for weight loss during lactation and for the overall cycle. The gain of backfat thickness during gestation of PFP sows was higher than for sows of other treatments (*p* = 0.02); PFP sows tended to lose more backfat thickness during lactation than sows of other treatments (*p* = 0.09). However, for the overall cycle, there was no difference in backfat thickness gain between treatments (*p* = 0.69). Feed intake during gestation and lactation did not differ between treatments. Litter weight at birth was higher for BF and PFP sows than for PFI sows, and FF sows were intermediate (*p* = 0.05). Birth-to-weaning mortality was lower for PFP and PFI sows than for BF sows, and FF sows were intermediate (*p* = 0.001). PFP sows weaned more piglets than the other treatments (*p* = 0.02), but no difference was observed in the weaning rate of the piglets between treatments. No difference in litter performance was observed between treatments considering total piglets born, number of piglets born alive, rate of stillborn piglets, average weight at 24 h after birth, total weight of litter at weaning, and weight gain of the litter or per piglet during lactation.

#### 3.1.2. Cycle 2

During the second cycle, there were minimal differences between feeding strategies for sow body condition or performance in lactation (Table 4). Live weight at breeding, after farrowing, and at weaning and weight gain in gestation, lactation, and for the overall cycle were not different between treatments. At farrowing, PFI sows had a higher backfat thickness than FF sows (BF and PFP sows were intermediate [*p* = 0.05]), and no differences were observed at breeding and at weaning. The gain of backfat thickness was significantly higher during gestation for PFI sows than for FF sows, but there was a tendency for PFI sows to have a greater mobilization during lactation (*p* = 0.10) than FF sows (BF and PFP sows were intermediate). Feed intake during gestation and lactation did not differ between treatments. There were no significant effects on litter performance during the second cycle for total born, number of born alive, rate of stillborn piglet, total weight of litter at birth, piglet average weight at birth, number or rate of weaned piglets, total litter weight at weaning, or piglet’s daily weight gain. Only two trends were observed: an increase in piglet birth-to-weaning mortality for PFP sows (*p* = 0.09) and a higher litter weight gain for PFP sows during lactation (*p* = 0.07).

#### 3.1.3. Cycle 3

During the third cycle, there was no observed effect of treatments on weight or backfat thickness at breeding, after farrowing, and at weaning (Table 5). Weight gain during gestation tended to be higher for PFP sows than for FF sows (*p* = 0.06), and backfat thickness gain during gestation also tended to be higher for PFI and PFP sows than for FF and BF sows (*p* = 0.08), with a higher loss of fat for PFI and PFP sows during lactation (*p* = 0.03). Stillborn rates were lower for PFP sows than for BF and FF sows (*p* = 0.01), with PFI sows having intermediate stillborn rates. No significant effects were observed on total born, number of born alive, total weight of litter at birth, piglet average weight at birth, piglet birth-to-weaning mortality, number or rate of weaned piglets, total litter weight at weaning, litter weight gain, and piglet daily weight gain. After three cycles, the number of batches skipped per sow, representing unproductive days, and the number of sows culled after each cycle did not differ between treatments (Table 6).

### 3.2. Effects on Plasma Calcium, Phosphorus, and 25(OH)D_3_

Plasma Ca was influenced by time of collection (*p* = 0.001; Table 7); plasma Ca was higher at the end of lactation than during gestation. Plasma P was not influenced by parity or treatment. A tendency for an interaction between parity and treatment (*p* = 0.09) was observed for plasma Ca (Figure 2), showing that animals in the PFI treatment group had lower plasma Ca in the first parity, regardless of the time of sampling.

### 3.3. Effects on Piglets After Weaning

No interactions between parity and treatment were observed, and the average performance of piglets over the three cycles is presented in Table 8. Unintentionally, the weights of piglets at weaning were higher for the FF treatment than for the BF and PFI treatments, which in turn were higher than PFP treatments (*p* < 0.001). The growth performance of the piglets was not affected by the maternal dietary treatments during the first 14 days post-weaning phase. During the second phase, piglets from sows receiving the FF treatment had higher ADG (*p* < 0.05) and a lower feed conversion ratio (FCR, *p* < 0.05) than the piglets from sows receiving the BF treatment, while piglets from sows receiving the PFP and PFI treatments were intermediate. With regard to ADFI, no difference was observed across treatments. In the third phase, piglets from sows receiving FF and PFP treatments had a higher ADG (*p* < 0.05) than those from the PFI treatments, while piglets from sows receiving the BF treatment were intermediate. Additionally, ADFI (*p* < 0.05) was higher for piglets from sows receiving the FF treatment than for piglets from sows receiving the BF and PFI treatments, and piglets from sows receiving the PFP treatment were intermediate. At 42 days, piglet weight was higher in the FF treatment than in the BF treatment, with PFP and PFI treatments being intermediate (*p* < 0.05).

On average across the three phases, piglets from sows in the FF treatment had higher ADG (*p* < 0.05) than those from the BF and PFI treatments, while piglets from sows receiving PFP during gestation were intermediate. The ADFI and FCR were similar across treatments. The BMC at day 14 tended to be lower in the PFI and PFP treatments (*p* = 0.07); at day 42, the BMC was 10% lower for the PFI treatment than for the FF treatment (*p* = 0.01). As a result, piglets from sows receiving the PFI treatment had a lower BMC gain (*p* < 0.05). Lean tissue was not different between treatments at day 14 and was 8% lower in the PFI treatment than in the FF treatment at day 42 (*p* = 0.01), with piglets from sows receiving BF and PFP treatments being intermediate. As a result, lean tissue gain was lower in the PFI treatment than in the FF treatment (*p* = 0.02), while other treatments were intermediate. There were no differences in fat deposition among treatments.

## 4. Discussion

### 4.1. Flat Feeding vs. Bump Feeding Strategies

The first objective of this project was to validate the impact of the bump feeding strategy on sow performance. This feeding strategy is widely adopted by commercial pig farmers because of its simplicity. Indeed, this strategy involves increasing feed intake at the end of gestation to meet the higher nutritional demands during this stage while still using only one diet for gestation [5]. Our study showed little or no effect of the bump feeding strategy on the body condition of sows or their reproductive performance, which is consistent with previous studies in gilts [7] or in sows across different cycles [5]. There was no significant difference in culling rate or unproductive days between BF and FF sows, although numerically, FF sows had fewer unproductive days than BF sows (10.1 vs. 16.0 d) and a lower number of culled gilts (7.3% vs. 12.4%). Regarding post-weaning performance of piglets, as there was a significant difference between treatments on BW at weaning that was not found in the overall population of piglets, it was introduced as a covariable in all the variables tested. Nevertheless, weight at the end of the post-weaning period was higher in the FF treatment, indicating a higher ADG than was achieved with the BF strategy.

The higher ADG for piglets in the FF treatment may be related to their higher weight, which could indicate a greater maturity at the beginning of the experiment, or because the FF treatment provided 6% more feed from days 0 to 89 of gestation and 17% less feed during late gestation than the BF treatment (Table 2). Muscle fibers develop during gestation, and the number of fibers is established at birth [27]. A reduction in nutritional intake during mid-gestation (35 to 55 days) may result in decreased muscle fiber hyperplasia, which may limit the number of fibers present in the piglets at birth. Indeed, lean content and lean gain during nursery were numerically higher for FF piglets when compared to BF piglets. Some studies have also shown that energy intake between days 25 and 50 of gestation can affect the performance of piglets [17]. Piglets that develop a higher number of muscle fibers during gestation can have enhanced growth potential after birth [28]. However, it is worth nothing that the FF piglets selected for the post-weaning study had a higher weight at the beginning of the post-weaning period which could have impacted the results. Also, the performance of these piglets was not followed during the growing–finishing phase to determine if the better performance was maintained. Additional studies are needed to validate those effects.

Overall, our study did not indicate any benefit of implementing a bump feeding strategy, and no major negative impacts were observed except for a lower ADG for post-weaning piglets. Because there are no additional feeding costs associated with either the flat or bump feeding strategy, the results of this study suggest that either strategy would have a similar effect on gilts and sows; however, the impact on post-weaning piglets requires further study.

### 4.2. Precision Feeding Strategies

The second objective of this project was to validate the impact of the precision feeding strategy on sow performance. Precision feeding reduced nitrogen intake by 10 and 13%, SID Lys intake by 13 and 17 % and total P intake by 6% and 9% for the PFP and PFI treatments, respectively, when compared with conventional treatments (average intake of BF and FF treatments). This result is consistent with a study that observed a similar decrease in P intake (−8%) but a higher decrease in SID Lys intake (−25%; 15). In another study, SID Lys intake was reduced by 16%, with no reduction on P intake [17].

Piglet mortality was reduced during cycles 1 and 3 with the precision feeding treatments. During the first cycle, PFP sows weaned significantly more piglets than sows from other treatments, probably because of the combined effect of having a significant decrease in birth-to-weaning piglet mortality and that cross-fostering was carried out between sows of the same dietary treatment. In cycle 2, few effects were observed during lactation. In the third cycle, PFP sows had significantly lower stillborn piglet rates than did sows from BF and FF treatments, with PFI sows being intermediate. A previous study on precision feeding during gestation conducted over two reproductive cycles also showed a trend towards reduced stillborn rates with precision feeding by parity [16]; however, other studies have observed no impact on piglet mortality [15,17]. When comparing the SID Lys intake between experiments, considering an average for all sows during gestation, the SID Lys intake per day for the precision feeding treatment was 10% and 15% higher in our study for the PFI and PFP treatments (respectively 11.9 and 11.5 g SID Lys intake/day) when compared with other studies (10.4 g SID Lys intake/day) [15,17]. The average amount of SID Lys intake per day in control or conventional treatments was like that observed by Gaillard and Dourmad [15] but was 11% higher than that observed by Stewart et al. [17]. Similar performances were observed for sows in terms of litter size and average piglet weight at birth. On average, sows receiving the PFP and PFI treatments consumed 17.6 g SID Lys per day between days 90 and 110 of gestation, whereas other studies used a lower SID Lys feeding rate during a similar stage of gestation (13.0 g SID Lys from days 79 to 108 of gestation [17] and 13.8 g SID Lys from days 89 to 110 of gestation [15]). The higher level of SID Lys supply per day but similar performance in lactation could explain the differences between these studies. Indeed, other studies testing different levels of Lys during gestation have also observed that providing 18.5 g of SID Lys per day throughout the entire gestation period reduced the stillborn rate in multiparous sows [29], that supplying 16.0 g/day during late gestation maximized the total number of piglets born alive [30], and that supplying 20 g of SID Lys per day to sows during late gestation when compared to 10.7 g SID Lys per day reduced preweaning mortality [10]. Mechanisms behind this reduction in mortality when feeding more lysine at the end of gestation for sows are, however, unclear.

At the end of gestation, conventional feeding has been shown to not fully meet the lysine digestible requirements of sows [18,19,20]. The end of gestation is characterized by rapid fetal growth and high protein deposition [31,32]. Some studies have shown that nutritional or energetic restrictions during gestation can negatively affect embryonic and postnatal survival of piglets [31,33]. Nevertheless, a tendency for hypocalcemia was observed in PFI sows at days 21 and 112 of gestation and at day 21 of lactation during cycle 1. Gilts are still growing during the first parity, which includes their bone [34]. The plasma Ca results indicate that the Ca level given to PFI sows was probably lower than their requirements as early as day 21 of gestation. During lactation, although plasma Ca was increased (Table 7) likely through a higher Ca intake, PFI sows still exhibited reduced calcemia. As Ca is highly regulated [35] and the demand for Ca increases during lactation [3], the hypocalcemia observed in the PFI sows indicates that the level of Ca in the lactation diet was limiting for PFI gilts. In a recent study, Lemée et al. [36] showed that precision feeding during gestation following the Bikker and Blok [24] models of requirements resulted in a lack of Ca to retain P into bone.

Precision feeding resulted in fluctuations in the body composition of sows within each cycle when compared with conventional feeding (Figure 3). As an example, PFP sows gained significantly more backfat during gestation in the first cycle but also lost more fat during lactation. Similarly, in the second cycle, PFI sows gained more fat during gestation but also tended to lose more fat during lactation. Similar results were obtained for both PFI and PFP treatments during the third cycle. One explanation is that precision feeding reduces the excess protein in the diet, which could promote more fat deposition during gestation. Excess protein must be metabolized, requiring energy; thus, by reducing excess protein, more energy is available for fat gain during gestation [37]. However, during lactation, higher mobilization of reserves happened, which could have led to the higher number of unproductive days for the sows receiving the PFP treatment [38]. Also, even if not significant, precision feeding treatments resulted in a numerical increase in the culling rate (+14.6%). However, Stewart et al. [17] did not observe any significant negative impacts on the number of sows removed from their study, and they actually found that 30% more sows were removed from the control treatment. There is no clear hypothesis to explain this higher fat mobilization for PFP and PFI sows. It is known that gaining too much fat or weight in gestation may negatively affect the lactation performance of sows [6,9], but we do not expect it to be the main mechanism since backfat thickness at farrowing was on target of the genetic objective [23,38]. Since PFP sows weaned more piglets, better performance in lactation with similar lactation feed intake could also explain the higher fat mobilization but only for gilts, as no difference in weaned performance was observed for second and third parity sows.

When comparing the two precision feeding strategies, sows receiving the PFP treatment weaned significantly more piglets and had a higher litter weight after 24 h in cycle 1 than sows receiving the PFI treatment, although the birth-to-weaning mortality rate was similar between the two treatments. In cycle 2, no difference was observed between the two treatments, and in cycle 3, the stillborn rate was lower for sows receiving the PFP treatment. Post-weaning piglets from PFP sows were the lighter at the beginning of the nursery phase but did not end up being significantly different from the FF piglets that had a higher ADG than the BF and PFI piglets. The PFI piglets tended to have a lower BMC at day 14, which was significant at day 42, resulting in lower BMC gain regardless of parity. These results showed that the PFP treatment had more pronounced positive effects on sows and piglets performance than the PFI treatment.

One of the main differences between these two treatments is the first third of gestation, where sows receiving the PFI strategy consumed 18%, 20%, and 14% less SID Lys, digestible phosphorus, and total calcium, respectively, than sows receiving the PFP strategy because of the adjustment to the actual sow body weight at weaning for PFI sows (Table 2). Results showed a decrease in plasma calcium in first-parity PFI sows at days 21 and 112 of gestation. As the Ca/P ratio of milk is considered fixed [39,40], the lower Ca status during gestation likely explains the lower BMC of piglets, at least during the first parity. Ca is highly regulated, so there may be regulatory mechanisms that maintain calcemia during parities 2 and 3, but this could still influence the piglets receiving the PFI treatment [41]. This will need to be confirmed with more data on piglet bone mineralization. Other studies testing individualized precision feeding also saw little effect on sow performance [15,17]. Therefore, the slightly higher performance of PFP sows can be related to the higher nutritional inputs provided by the PFP treatment early in gestation, which promotes better fetal and placental development than for PFI. Notably, early gestation has been less frequently studied than the last third of gestation, where it is well known that dietary requirements increase exponentially.

### 4.3. Implementation

Applying precision feeding during gestation requires knowledge of the historical performance of the herd to establish nutritional requirement curves. Models such as INRAporc or NRC require established parameters such as sow weight, parity, litter size, average piglet weight, and target weight and fat gain for each sow [19,25]. Although it is possible to individualize nutritional inputs for each sow and to know the actual parameters for each sow (e.g., parity and weight at breeding), several parameters remain poorly estimated (e.g., average piglet weight or litter size). Therefore, a safety margin may be needed to avoid underestimating the sow nutritional requirements. It is also important to know the basis of the model being used, specifically whether it estimates the requirements of an individual animal or the requirements of a group of individuals, meaning that some safety margins may already be included [42]. Establishing the safety margin may involve simply increasing supplies by a percentage (e.g., +10 to 15%) or by choosing performance parameters that are higher than the average, as we have done in this project. The parameters used for the PFP treatment in this study were the average performance values with the addition or subtraction of a standard deviation to avoid underestimating the nutritional requirements of most sows of the same parity; this likely explains the higher SID Lys intakes than those used in similar studies [15,17]. Our parameters were established based on previous studies with similar genetic and sanitary status and were used because we had no historical data from the research farm used in this study [16,21]. This lack of historical data is a limitation for precision feeding implementation. Nevertheless, in the current study, comparing the observed performance with expected performance indicates that the parameters to the established requirement would have been slightly higher than those initially used (approximately 8%). Another limitation of our study was that we used a new herd at a new farm, which posed some challenges, including health issues during the first cycle. A first-parity sow entering a new farm previously housing only gilts may be more stressed than it would be at a farm that was already housing older sows. This stressful environment may cause differences in comportment, behavior, temperament, and, possibly, nutritional requirements.

This article is a revised and expanded version of two papers entitled “Impact de l’alimentation de précision et d’un surplus d’aliment en fin de gestation sur les performances et l’état corporel de truies suivies pendant leurs deux premiers cycles de gestation et lactation” [43], which was presented at Journées de la recherche porcine, France, February 2023, and “Impact de l’alimentation de précision en gestation sur la carrière productive de truies suivies pendant trois cycles reproductifs” [44], which was presented at Journées de la recherche porcine, France, February 2024.

## 5. Conclusions

The use of the bump feeding strategy, which involves redistributing energy intake during gestation to better fit requirements, did not significantly improve sow performance during lactation. Considering post-weaning piglet performance, the flat feeding strategy appeared to enhance the ADG of piglets when compared with the bump feeding strategy; however, these results should be confirmed with further studies. The current study demonstrated that as precision feeding did not differ from flat feeding, it is possible to reduce nitrogen intake by 10–13% and total phosphorus intake by 6–9%. Additionally, precision feeding reduced piglet mortality during lactation for gilts and third-parity sows. The effect was more pronounced with PFP than PFI, which might be due to the lower intake of SID Lys, digestible phosphorus, and total calcium for PFI during the first 28 days of gestation. More research on nutrient requirements during early gestation is needed to optimize the nutrient supply during this period. Based on the results of this study, the precision feeding per parity strategy is preferred for commercial implementation over individualized precision feeding. When implementing precision feeding during gestation, the parameters used to estimate nutrient requirement curves should be carefully selected considering the historical data of the herd and should include safety margins to not underestimate the nutritional requirements of sows.

## Figures and Tables

**Figure 1 animals-14-03513-f001:**
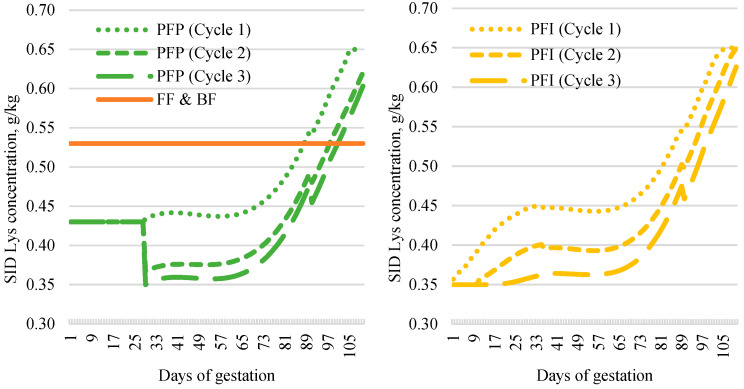
Standardized ileal digestible lysine (SID Lys) content of feeds distributed according to dietary treatment (flat feeding, FF; bump feeding, BF; precision feeding per parity, PFP; and precision feeding per individual, PFI), gestation day, and parity.

**Figure 2 animals-14-03513-f002:**
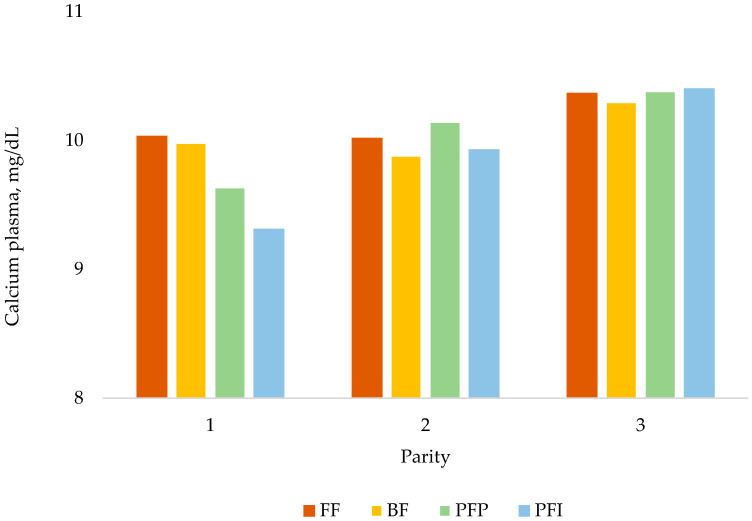
Plasma calcium as a function of treatment and parity. Treatment, *p* = 0.61; parity, *p* < 0.001; treatment × time, *p* = 0.09 (flat feeding, FF; bump feeding, BF; precision feeding per parity, PFP; and precision feeding per individual, PFI).

**Figure 3 animals-14-03513-f003:**
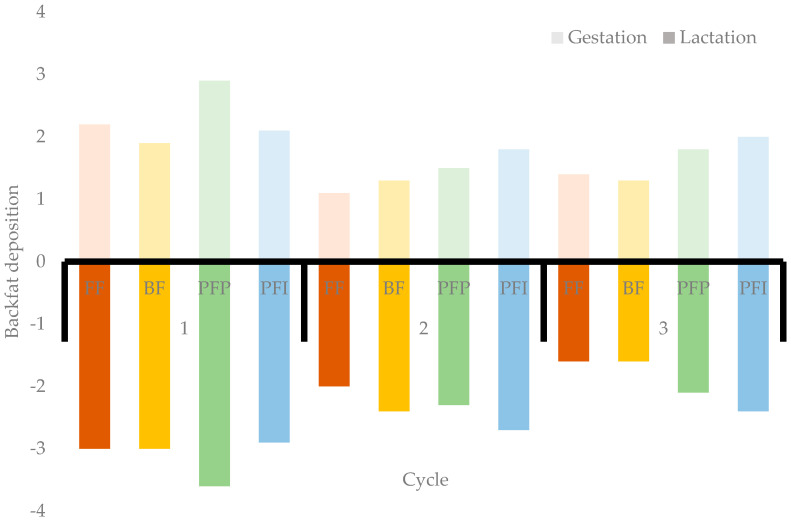
Backfat thickness gain in gestation and lactation according to dietary treatments during gestation over 3 cycles.

**Table 1 animals-14-03513-t001:** Ingredient and nutritional composition of experimental feeds.

Ingredient g/kg	Feed A	Feed B	Lactation
Corn	578	423	614
Soybean meal—47%	-	39.1	231
Wheat	-	-	50
Wheat bran	300	300	-
Corn distillers grains (Varennes)	-	139	50.0
Oat hulls	101	37.2	-
Canola meal	-	25.0	-
Animal fat	-	-	14.2
Limestone	11.6	22.4	14.0
Monocalcium phosphate	-	1.92	10.2
Salt	4.93	4.49	4.80
Lysine HCl	0.950	2.50	3.70
D.L. Methionine	-	-	1.20
L-threonine	0.290	1.12	1.30
L-tryptophane	-	-	0.200
L-valine	-	-	0.900
Choline chloride	0.720	0.720	1.15
Phytase ^1^	0.0400	0.150	0.150
Sow micro-premix ^2^	2.50	2.50	2.50
Liquiprop liquid ^3^	0.50	0.50	-
**Nutritional content calculated (analyzed) ^4^**			
Net energy sow, kcal/kg	2220	2219	2540
Crude protein, %	10.0 (10.2)	15.4 (16.2)	17.4 (18.0)
Total lysine, %	0.45 (0.45)	0.81 (0.80)	1.15 (1.15)
Total methionine, %	0.16 (0.16)	0.31 (0.27)	0.36 (0.35)
Total methionine + cysteine, %	0.38 (0.36)	0.61 (0.56)	0.60 (0.63)
Total threonine, %	0.36 (0.37)	0.66 (0.64)	0.74 (0.77)
Total tryptophan, %	0.11 (0.13)	0.16 (0.19)	0.21 (0.22)
Total isoleucine, %	0.32 (0.32)	0.56 (0.54)	0.67 (0.70)
Total valine, %	0.45 (0.45)	0.73 (0.71)	0.83 (0.87)
Total leucine, %	0.79 (0.84)	1.39 (1.41)	1.46 (1.52)
Total arginine, %	0.59 (0.55)	0.90 (0.86)	1.01 (1.05)
SID lysine, % ^5^	0.35	0.65	1.01
SID methionine, %	0.14	0.26	0.25
SID methionine + cysteine, %	0.31	0.50	0.50
SID threonine, %	0.28	0.52	0.53
SID tryptophan, %	0.08	0.13	0.15
SID isoleucine, %	0.26	0.43	0.62
SID valine, %	0.37	0.56	0.75
SID leucine, %	0.71	1.11	1.33
SID arginine, %	0.50	0.75	0.99
Fat, %	3.30	4.10	3.85
Crude fiber, %	7.10	6.10	2.62
Total calcium, %	0.52 (0.67)	1.00 (1.10)	8.1 (9.2)
Phosphorus, %	0.47 (0.51)	0.64 (0.69)	0.58 (0.60)
Apparent digestible phosphorus, %	0.20	0.40	0.60

^1^ Quantum B 5000 L, ABvista, Marlborough, United Kingdom. ^2^ Premix formulations per kg: 4000 IU vitamin A acetate, 600 IU vitamin D3, 24 IU vitamin E (dl-α-tocopheryl acetate), 1200 mg menadione (niacin), 8 mg vitamin B12, 600 mg thiamine mononitrate, 2400 mg riboflavin, 10,000 mg calcium D-pantothenate, 12,000 mg niacin/niacinamide, 1200 mg pyridoxine, 320 mg biotin, 3200 mg folic acid, 80 mg chromium (chromium propionate), 16,000 mg manganese (MnSO_4_), 40,000 mg iron (FeSO_4_), 6000 mg copper (CuSO_4_·5H_2_O), 50,000 mg zinc (ZnSO_4_), 200 mg iodine (EDDI C2H10I2N2), 40 mg selenium (Na_2_SeO_2_), 80 mg selenium (hydroxyanalogue of selenomethionine). ^3^ Anti-mold, Agro-Bio Contrôle Inc., Saint-Hyacinthe, QC, Canada. ^4^ Calculation are based on NRC, 2012 [25] ^5^ Standardized ileal digestible.

**Table 2 animals-14-03513-t002:** Gestation feed intake, proportion of feed A and B, and standardized ileal digestible lysine (SID Lys), apparent digestible phosphorus (P dig), and total calcium (Ca) intake according to dietary treatments during gestation over 3 cycles.

Cycle	TRT	Feed Intake kg/j				Feed A % ^1^				SID Lys Intake g/d ^2^				Digestible P Intake g/d				Total Ca Intake g/d			
		1-28 d	29-89 d	90-110 d	Global	1-28 d	29-89 d	90-110 d	Global	1-28 d	29-89 d	90-110 d	Global	1-28 d	29-89 d	90-110 d	Global	1-28 d	29-89 d	90-110 d	Global
1	FF	2.66	2.5	2.49	2.54	40	40	40	40	14.1	13.3	13.2	13.5	8.5	8.0	8.0	8.1	24.7	23.3	23.2	23.6
	BF	2.6	2.3	2.98	2.50	40	40	40	40	13.8	12.2	15.8	13.3	8.3	7.4	9.5	8.0	24.2	21.4	27.7	23.3
	PFP	2.68	2.31	2.99	2.53	66	64	17	56	12.1	10.6	17.9	12.4	7.2	6.2	11.1	7.4	22.0	18.9	30.8	21.9
	PFI	2.65	2.3	2.98	2.52	80	64	15	59	10.9	10.6	18.2	12.1	6.4	6.2	11.0	7.1	20.1	18.9	31.0	21.4
2	FF	2.9	2.63	2.57	2.69	40	40	40	40	15.4	13.9	13.6	14.3	9.3	8.4	8.2	8.6	27.0	24.5	23.9	25.0
	BF	2.75	2.49	3.07	2.66	40	40	40	40	14.6	13.2	16.3	14.1	8.8	8.0	9.8	8.5	25.6	23.2	28.6	24.8
	PFP	2.82	2.51	3.08	2.70	69	79	25	66	12.4	10.3	17.9	12.2	7.3	6.0	10.8	7.2	22.6	19.1	30.5	22.1
	PFI	2.75	2.5	3.08	2.67	89	77	23	70	10.5	10.5	17.9	11.9	6.1	6.3	10.8	7.0	19.8	19.3	30.8	21.5
3	FF	2.98	2.66	2.58	2.73	40	39	39	39	15.8	14.1	13.7	14.5	9.5	8.5	8.3	8.7	27.7	24.7	24.0	25.4
	BF	2.83	2.53	3.12	2.72	40	40	40	40	15.0	13.4	16.5	14.4	9.1	8.1	10.0	8.7	26.3	23.5	29.0	25.3
	PFP	2.88	2.54	3.12	2.74	69	89	37	74	12.7	9.7	16.8	11.8	7.5	5.6	10.3	7.0	23.0	18.3	29.3	21.6
	PFI	2.84	2.52	3.12	2.71	96	89	37	81	10.2	9.6	16.8	11.1	6.0	5.5	10.3	6.5	19.6	18.1	29.3	20.6

^1^ FF and BF sow received a fixed blend of feed A and feed B throughout gestation. PFP received a fixed blend from 1 to 28 days but a daily blend matching the requirement curves for the rest of gestation. PFI received a daily blend matching the requirement curve throughout gestation. ^2^ SID = standardized ileal digestible; BF, bump feeding; FF, flat feeding; PFP, precision feeding per parity; PFI, precision feeding individual.

**Table 3 animals-14-03513-t003:** Sow body composition and performance in lactation during the first cycle.

Variables, Units	FF		BF		PFP		PFI		SEM ^2^	*p* Value
	N ^1^	Av.	N	Av.	N	Av.	N	Av.		
Sows										
Weight at breeding, kg	119	150.2	125	150.9	123	148.7	129	150.5	1.2	0.32
Weight at 110 days of gestation, kg	103	213.7	108	215.3	106	216.5	110	215.6	1.4	0.27
Weight at weaning, kg	122	186.2	125	185.1	122	185.5	127	187.0	1.7	0.72
Weight gain in gestation, kg	99	62.9 ^b^	106	63.9 ^b^	103	66.7 ^a^	109	64.5 ^b^	1.1	0.01
Weight gain in lactation, kg	103	−27.3	107	−30.2	104	−30.5	108	−28.8	1.5	0.14
Overall weight gain cycle 1, kg	99	35.6	105	33.6	101	36.1	108	35.7	1.6	0.41
Backfat thickness (BT) at breeding, mm	121	13.7	126	14.3	123	13.6	128	14.1	0.4	0.25
BT at 110 days of gestation, mm	104	16.2	108	16.4	103	16.6	109	16.5	0.5	0.85
BT at weaning, mm	122	13.2	125	13.4	121	13.1	129	13.4	0.4	0.74
BT gain in gestation, mm	102	2.2 ^b^	106	1.9 ^b^	100	2.9 ^a^	107	2.1 ^b^	0.3	0.02
BT gain in lactation, mm	104	−3.0 ^a^	107	−3.0 ^a^	101	−3.6 ^b^	109	−2.9 ^a^	0.3	0.09
Overall BT gain cycle 1, kg	102	−0.7	105	−1.1	98	−0.6	107	−0.8	0.4	0.69
Total feed intake in gestation, kg	121	276.0	127	273.7	123	276.5	128	274.5	4.4	0.46
Feed intake in lactation, kg/day	119	5.4	124	5.4	120	5.3	125	5.4	0.1	0.89
Litters										
Total piglet born, n	122	14.41	127	14.69	124	14.74	129	14.36	0.38	0.67
Piglet born alive, n	122	13.66	127	13.9	124	13.80	129	13.63	0.37	0.88
Stillborn rate, % total piglet born ^3^	122	5.1	127	4.9	124	6.1	129	5.3		0.30
Litter birth weight, kg	122	18.55 ^ab^	126	19.07 ^a^	124	19.03 ^a^	129	18.00 ^b^	0.44	0.05
Piglet birth weight, kg	122	1.31	126	1.32	124	1.31	129	1.27	0.02	0.18
Birth to weaning mortality, % ^3^	122	11.3 ^ab^	127	13.8 ^a^	124	8.7 ^b^	129	10.3 ^b^		0.001
Weaned piglets, n	122	12.0 ^b^	137	11.9 ^b^	124	12.5 ^a^	129	11.8 ^b^	0.2	0.02
Weaned piglets, % piglets after cross-fostering	122	92.2	137	91.9	124	94.4	129	92.2		0.18
Litter weaning weight, kg	119	62.6	125	63.3	119	65.4	129	63.1	1.6	0.30
Litter weight gain, kg	114	46.8	114	48.2	117	49.2	123	48.5	1.3	0.32
Average daily gain per piglet, g/d	114	219.8	114	224.4	117	217.5	123	223.7	4.0	0.25

^1^ BF, bump feeding; FF, flat feeding; N = number of sows; PFP, precision feeding per parity; PFI, precision feeding individual. ^2^ Standard error of the means (maximum value); analysis of variance with treatment as the main effect and sow or litter as the experimental unit; multiple comparisons, with different letters on the same line indicating a significant difference (*p* < 0.05). ^3^ Logistic regression.

**Table 4 animals-14-03513-t004:** Sow body composition and performance in lactation during the second cycle.

Variables, Units	FF		BF		PFP		PFI		SEM ^2^	*p* Value
	N ^1^	Av.	N	Av.	N	Av.	N	Av.		
Sows										
Weight at breeding, kg	95	186.9	82	186.7	75	184.9	84	185.5	1.68	0.72
Weight at 110 days of gestation, kg	95	243.5	82	242.9	75	243.4	84	244.6	2.04	0.85
Weight at weaning, kg	94	217.9	82	216.6	76	216.3	84	215.9	2.39	0.84
Weight gain in gestation, kg	95	56.4	82	56.3	75	58.2	84	59.3	1.63	0.14
Weight gain in lactation, kg	93	−25.7	80	−26.1	75	−27.1	84	−28.7	1.91	0.37
Overall weight gain, kg	93	30.6	80	29.8	75	31.1	84	30.6	1.4	0.86
Backfat thickness (BT) at breeding, mm	94	13.3	78	14.1	78	13.5	84	13.8	0.5	0.37
BT at 110 days of gestation, mm	94	14.6 ^b^	77	15.3 ^ab^	78	14.9 ^ab^	84	15.5 ^a^	0.3	0.05
BT at weaning, mm	94	12.6	78	13.1	76	12.5	84	12.9	0.3	0.98
BT gain in gestation, mm	95	1.1 ^b^	82	1.3 ^ab^	74	1.5 ^ab^	84	1.8 ^a^	0.3	0.04
BT gain in lactation, mm	95	−2.0	81	−2.4	75	−2.3	84	−2.7	0.3	0.10
Overall BT gain, kg	95	−1.0	81	−1.0	74	−0.8	84	−0.9	0.3	0.89
Total feed intake in gestation, kg	92	298.0	83	294.3	75	299.2	84	297.2	2.19	0.22
Feed intake in lactation, kg/day	87	7.1	78	7.2	67	7.2	77	6.9	0.2	0.12
Litters										
Total piglet born, n	96	13.65	84	13.83	76	14.16	84	14.20	0.58	0.70
Piglet born alive, n	96	12.92	84	13.11	76	13.2	84	13.44	0.56	0.80
Stillborn rate, % total piglet born ^3^	96	4.9	84	5.3	76	6.1	84	4.8		0.36
Litter birth weight, kg	96	19.89	84	19.93	76	20.15	84	19.85	0.66	0.54
Piglet birth weight, kg	96	1.50	84	1.49	76	1.47	84	1.42	0.04	0.15
Birth to weaning mortality, % ^3^	96	8.4	84	8.1	76	10.9	84	9.8		0.09
Weaned piglets, n	96	11.52	84	11.98	76	11.78	84	11.85	0.33	0.51
Weaned piglets, % piglets after cross-fostering	96	93.3	84	95.3	76	93.4	84	93.6		0.61
Litter weaning weight, kg	96	73.6	84	77.4	76	76.2	84	75.6	2.3	0.35
Litter weight gain, kg	92	56.3	83	59.9	72	60.5	81	58.9	1.8	0.07
Average daily gain per piglet, g/d	91	257.9	82	264.0	72	262.0	81	262.9	5.8	0.70

^1^ BF, bump feeding; FF, flat feeding; N = number of sows; PFP, precision feeding per parity; PFI, precision feeding individual. ^2^ Standard error of the means (maximum value); analysis of variance with treatment as the main effect and sow or litter as the experimental unit; multiple comparisons, with different letters on the same line indicating a significant difference (*p* < 0.05). ^3^ Logistic regression.

**Table 5 animals-14-03513-t005:** Sow body composition and performance in lactation during the third cycle.

Variables, Units	FF		BF		PFP		PFI		SEM ^2^	*p* Value
	N ^1^	Av.	N	Av.	N	Av.	N	Av.		
Sows										
Weight at breeding, kg	78	218.7	75	216.4	60	215.7	68	215.05	1.7	0.43
Weight at 110 days of gestation, kg	76	258.4	74	258.5	58	259.4	66	257.5	1.7	0.90
Weight at weaning, kg	75	235.2	72	233.1	60	232.8	68	230.1	1.9	0.30
Weight gain in gestation, kg	75	39.4 ^b^	73	42.5 ^a^	58	44.0 ^a^	66	42.6 ^a^	1.5	0.06
Weight gain in lactation, kg	73	−23.5	72	−25.2	58	−26.4	66	−27.6	2.3	0.18
Overall weight gain, kg	73	16.0	72	17.5	60	17.6	67	14.8	1.8	0.31
Backfat thickness (BT) at breeding, mm	78	12.6	75	13.2	60	12.4	68	12.6	0.4	0.41
BT at 110 days of gestation, mm	76	14.0	74	14.4	58	14.1	66	14.5	0.4	0.71
BT at weaning, mm	76	12.4	74	12.8	58	12.0	65	12.2	0.4	0.43
BT gain in gestation, mm	76	1.4	74	1.3	58	1.8	66	2.0	0.3	0.08
BT gain in lactation, mm	76	−1.6 ^a^	74	−1.6 ^a^	58	−2.1 ^b^	65	−2.4 ^b^	0.3	0.03
Overall BT gain, kg	76	−0.3	74	−0.3	58	−0.3	65	−0.4	0.3	0.96
Total feed intake in gestation, kg	75	305.2	71	303.3	54	304.7	65	305.0	1.8	0.86
Feed intake in lactation, kg/day	75	7.3	72	7.5	58	7.3	64	7.2	0.2	0.42
Litter										
Total piglet born, n	76	15.4	74	15.6	58	15.1	66	15.6	0.4	0.83
Piglet born alive, n	76	13.9	74	14.1	58	14.1	66	14.4	0.4	0.89
Stillborn rate, % total piglet born ^3^	76	9.1 ^a^	74	10.4 ^a^	58	6.2 ^b^	66	7.0 ^ab^		0.01
Litter birth weight, kg	76	21.8	74	21.5	58	21.0	66	21.0	0.8	0.62
Piglet birth weight, kg	76	1.4	74	1.4	58	1.4	66	1.4	0.04	0.29
Birth to weaning mortality, % ^3^	76	13.2	74	12.5	58	13.7	66	11.3		0.76
Weaned piglets, n	76	12.3	74	12.4	58	12.5	66	12.7	0.3	0.68
Weaned piglets, % piglets after cross-fostering	76	91.2	74	92.5	58	92.9	66	92.6		0.83
Litter weaning weight, kg	76	76.1	74	76.9	58	78.7	66	78.8	2.3	0.66
Litter weight gain, kg	75	58.6	71	59.7	57	60.8	64	62.1	1.5	0.35
Average daily gain per piglet, g/d	75	237.8	69	246.6	57	248.1	64	246.9	4.0	0.19

1 BF, bump feeding; FF, flat feeding; N = number of sows; PFP, precision feeding per parity; PFI, precision feeding individual. ^2^ Standard error of the means (maximum value); analysis of variance with treatment as the main effect and sow or litter as the experimental unit; multiple comparisons, with different letters on the same line indicating a significant difference (*p* < 0.05). ^3^ Logistic regression.

**Table 6 animals-14-03513-t006:** Number of batches skipped and cumulative culled rate after each cycle per treatment.

Variables, Units	FF		BF		PFP		PFI		SEM ^2^	*p* Value
	N ^1^	Av.	N	Av.	N	Av.	N	Av.		
Number of batches skipped per sow after 3 cycles (unproductive days considering 28 days/batch)	97	0.36 (10.1)	103	0.57 (16.0)	92	0.67 (18.8)	99	0.61 (17.1)	0.14	0.19
Culling rate after first cycle, %	122	7.3	127	12.4	124	13.6	129	16.1	0.3	0.37
Cumulative culling rate after second cycle (variation with previous cycle), %	122	20.5 (+13.2)	127	18.9 (+6.5)	124	25.8 (+12.2)	129	23.3 (+7.2)	0.2	0.62
Cumulative culling rate after third cycle (variation with previous cycle), %	122	32.8 (+12.3)	127	30.7 (+11.8)	124	37.9 (+12.1)	129	34.9 (+11.6)	0.2	0.70

^1^ BF, Bump feeding; FF, flat feeding; N = number of sows; PFP, precision feeding per parity; PFI, precision feeding individual. ^2^ Standard error of the means (maximum value); analysis of variance with treatment as the main effect and sow or litter as the experimental unit; multiple comparisons, with different letters on the same line indicating a significant difference (*p* < 0.05).

**Table 7 animals-14-03513-t007:** Plasma concentrations of calcium, phosphorus, and 25(OH)D_3_ during gestation and lactation in function of time, treatment, and parity in sows.

	Calcium, mg/dL			Phosphorus, mg/dL			25(OH)D_3_, ng/mL	
	*21 d Gest*	*112 d Gest*	*21 d Lact*	*21 d Gest*	*112 d Gest*	*21 d Lact*	*112 d Gest*	*21 d Lact*
**Parity 1**								
FF	10.3	9.7	10.1	12.1	11.6	10.8	63.2	86.1
BF	9.9	9.7	10.4	11.6	11.4	10.7	53.3	86.6
PFP	9.4	9.7	9.9	11.4	11.7	9.6	64.2	70.4
PFI	9.3	9.1	9.6	12.2	11.3	10.2	58.8	89.8
**Parity 2**								
FF	9.9	10.0	10.0	11.6	12.1	11.7	42.5	48.5
BF	10.0	9.6	10.3	11.9	12.1	10.2	25.6	26.8
PFP	9.9	10.0	10.8	11.4	12.3	10.9	23.1	24.4
PFI	9.9	9.8	9.9	11.4	11.8	10.7	23.5	43.0
**Parity 3**								
FF	10.3	10.1	10.8	10.8	10.7	10.2	56.6	68.6
BF	10.1	10.1	10.8	10.6	10.6	10.7	48.2	62.1
PFP	10.3	10.0	10.8	10.4	10.8	10.5	52.8	68.0
PFI	10.7	10.0	10.4	10.1	11.1	9.8	49.9	67.7
***p*-value**								
Parity	0.614			0.472			0.037	
Time	0.001			<0.001			<0.001	
Treatment	0.0016			0.542			0.062	
Parity × Time	NS			<0.001			0.066	
Parity × Treatment	0.098			0.74			NS	
Time × Treatment	NS			NS			NS	
Parity · Time × Treatment	NS			NS			NS	

BF, bump feeding; FF, flat feeding; NS = non-significant; PFP, precision feeding per parity; PFI, precision feeding individual.

**Table 8 animals-14-03513-t008:** Average post-weaning performance and body composition of piglets from sows of cycles 1, 2, and 3 receiving different maternal feeding treatments during gestation.

Treatment	FF	BF	PFP	PFI	SEM ^1^	*p* Value
Live weight kg						
Day 0	6.3 ^a^	6.0 ^b^	5.8 ^c^	6.0 ^b^	0.04	0.001
Day 42	30.8 ^a^	28.9 ^b^	29.6 ^ab^	29.2 ^ab^	0.4	0.02
Average daily gain g/d						
0–14 days	332	312	321	323	12	0.70
14–28 days	696 ^a^	636 ^b^	647 ^ab^	657 ^ab^	20	0.05
28–42 days	791 ^a^	748 ^ab^	783 ^a^	738 ^b^	15	0.04
Global	601 ^a^	561 ^b^	576 ^ab^	567 ^b^	11	0.04
Average daily feed intake g/d						
0–14 days	388	374	372	379	10	0.64
14–28 days	833	799	796	797	21	0.51
28–42 days	1327 ^a^	1256 ^b^	1261 ^ab^	1255 ^b^	23	0.05
Global	847	814	808	809	15	0.21
Feed conversion ratio						
0–14 days	1.21	1.24	1.18	1.22	0.03	0.44
14–28 days	1.21 ^a^	1.27 ^b^	1.23 ^ab^	1.24 ^ab^	0.02	0.04
28–42 days	1.67	1.68	1.61	1.69	0.03	0.30
Global	1.41	1.45	1.40	1.42	0.02	0.28
Body composition (14–42 days)						
Bone mineral content (BMC) 14 d, g	169.8	167.4	157.4	157.1	3.69	0.07
BMC 42 d, g	413 ^a^	399 ^ab^	384 ^ab^	372 ^b^	7.93	0.01
BMC gain, g/j	7.94 ^a^	7.63 ^ab^	7.38 ^ab^	6.99 ^b^	0.25	0.05
Lean 14 d, g	8928	8261	8004	7850	2107	0.39
Lean 42 d, g	25917 ^a^	24498 ^ab^	24230 ^ab^	23198 ^b^	477	0.01
Lean gain, g/j	554 ^a^	537 ^ab^	529 ^ab^	500 ^b^	17.3	0.02
Fat gain, g/j	104	102	99.3	94.4	4.4	0.51

^1^ Standard error of the means (maximum value); analysis of variance with treatment as the main effect and sow or litter as the experimental unit; multiple comparisons, with different letters on the same line indicating a significant difference (*p* < 0.05).

## Data Availability

Data are unavailable due to privacy restrictions.

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
