# Peer review of "Impact of Precision Feeding During Gestation on the Performance of Sows over Three Cycles†"

_animals, 2024, doi:10.3390/ani14233513_

Round 1

Reviewer 1 Report

Comments and Suggestions for Authors

The manuscript by Cloutier et al. presents valuable research results on evaluating the effects of precision feeding during gestation on the performance of sows and their piglets over three reproductive cycles. These findings will be of significant interest to swine nutritionists and producers who are considering precision feeding in gestating sows to improve pig productivity.

Major Comments:

Specific Comments:

Abstract:

Line 32: Please describe the observations during the second cycle.

Line 35: Is there a reason why the bump feeding strategy focuses more on sow performance than piglet performance during lactation?

Introduction:

Line 47: Please consider “parity” as an individual characteristic.

Line 61-62: How was the first method of bump feeding on multiparous sows?

Line 64: Are the observed interesting effects from Quiniou [9]? Please add more details.

Line 83: Although the hypothesis focuses on gilts, this study investigates sow performance over three parities. The hypothesis needs to be revised.

Materials & Methods:

Line 102-105: There is insufficient information on feeding amounts by parity.

Line 153: Repeated statement “at the P2 position” and “at the last rib.”

Line 155-162: There are hyphenated words “artifi-cial,” “in-duced,” “lac-tation,” “per-formed.” Please revise these formatting errors throughout the manuscript.

Line 173: 3-phase feeding program; I could not find information on this 3-phase feeding program for weaned pigs (diet formulation or nutrient composition).

Line 238: Could the authors describe the changes in the number of sows (Table 3)? Why did the number of sows for BW and BT differ? Why did the number of sows for weight at weaning, weight gain in gestation, and weight gain in lactation differ?

Results:

Line 332: It is not ideal to use an abbreviation (ADFI) to start a sentence.

Discussion:

Line 371-374:

1. Greater maturity at the beginning of the experiment.” The authors have discussed muscle fiber hyperplasia, contributing to the maturity of the piglets. Considering the body composition result (Table 7) in the discussion would be helpful.

2. FF treatment provided 6% more feed from days 0 to 89 of gestation and 17% less feed during late gestation than the BF treatment.” For this discussion, the described references and discussions were not enough to support that statement.

3. Could it be that a limitation of this study or discussion is that muscle fiber development was not measured at weaning and 42 days post-weaning?

Line 419-424: The discussion is not enough to support your explanation for lower piglet mortality. To support your argument, the authors need to properly cite peer-reviewed papers showing that higher lysine intake during the gestation period reduces piglet mortality or stillborn rates compared with the current study’s lysine intake.

Line 429: [29,30]

Line 434: [31]

Line 450-454: It seems that the discussion is incomplete. Do the authors think that precision feeding treatments increase the culling rate because of fat mobilization while preventing excess protein intake (fat gain during gestation and more fat loss during lactation)?

Line 463: What is the meaning of “smallest”?

Line 458-472: I think the authors should clarify the purpose of this discussion more. It is not clear why they are having this discussion. For example, is this discussion for Ca and BMC results? Is this discussion for piglet performance?

Line 479-483: I believe there are some great studies that can be referred to in these statements. The authors might want to cite some references to support these statements.

Conclusion

Line 524-528: This part should probably be mentioned with greater detail in the discussion section rather than in the conclusion section.

Tables:

Table 1:

Lysine HCL, D.L. METHIONINE: The name/format of ingredients should be revised.

Please add manufacturer information for phytase and liquiprop liquid (anti-mold).

Please add a spec for sow micro-premix (vitamin, mineral premix spec).

Table 2: What are the meanings of “, comma” in the table? Are they decimal points?

Table 3: The format of superscripts “a,b” is incomplete. Please check the superscript format.

Some values use “. Point”, but some values use “, comma.” Please check and revise them throughout the tables.

Table 5: Some p-values have one decimal point (0.9, 0.3).

In the line of “weight gain in gestation,” superscript “a” was used for the lower value, and “b” was used for the higher values. In other variables, the superscript “a” was used for the higher value, and “b” was used for the lower value. Please check and revise them.

Figure 2 and 3: The label for the Y-axis overlaps with the numbers. Also, “,” should be revised to “.”.

Figure 2: Please indicate the meaning of superscripts “a,b.”

Table 7: What is the difference between using a “comma ,” and a “point .”? Please see the “bone mineral content” and “BMC gain.”

Reviewer 2 Report

Comments and Suggestions for Authors

Dear Editor,

The manuscript entitled Impact of precision feeding during gestation on the performances of sows and their piglets over three cycles, by Laetitia Cloutier et al. evaluated the impact of precision feeding and bump feeding strategies during gestation on the reproductive performances of sows monitored over three cycles. Their results showed that precision feeding reduce nitrogen intake and total phosphorus intake, while also decreasing piglet mortality during lactation. These findings suggest that the precision feeding per parity strategy maybe preferred for commercial implementation over individualized precision feeding. The English writing is good. Authors need to concern more the comments as followed.

General comments

1.      Figure 2 showed incomplete information. The treatment information of plasma calcium and phosphorus concentration was missed. Table (like Table 3) was more appropriate.

2.      It is not appropriate to use Figure to illustrate the plasma calcium as a function of treatment and parity. Also, table (like Table 3) was more appropriate.

3.      The data in Figure 4 had been shown in Table 3, 4, 5. This is a repeated display of data

Specific comments

1.      Many conjunctions are redundant. Such as, Line 158 “in-duced”, Line 159 “lac-tion”, Line 196 “in-troduced”.

2.      Line 429: why the sows in PRI group at days 21 and 112 of gestation during cycle1 was hypocalcemia? Just because their plasma calcium was lower than that at day 21 of lactation?

3.      Table 1: Provide the ingredient and nutritional composition of sow micro-premix

4.      Table 1: How is Vitamine D calculated

5.      Table 3: Some superscripts in Table 3 were not standardized.

6.      Check the format of Reference. As the journal name format was shown in both full journal name and abbreviation.

Reviewer 3 Report

Comments and Suggestions for Authors

The paper reported findings of a study evaluating the effects of bump feeding and precision feeding compared to conventional feeding on sow reproductive performance across three gestation and lactation cycles. The results showed that bump feeding did not significantly improve sow performance during lactation. However, precision feeding reduced nitrogen intake by 10-13% and phosphorus intake by 6-9%, while also decreasing piglet mortality during lactation. The paper has merit to be published at Animals, however, it needs some major adjustments related to structure and writing, before it is in an acceptable format. Some major and specific comments are provided below:

Main comments

-       The authors should attempt to give the introduction section a better flow of ideas. For example: “The first bump feeding strategy increases feeding costs [6,7,8], but the second strategy does not [9]. Studies have shown that the first bump feeding strategy has few benefits [8] and may have negative effects: sows may have excessive body conditions, which negatively impacts the farrowing process in multiparous sows [6,10] and the feed intake of gilts during lactation [6,7,11]. However, some studies using the first method of bump feeding have observed a positive effect on piglet birth-61 weight in gilts [10]. “ Such section could be narrowed down and written with a better connection between sentences/concepts.

-       I would also recommend that the authors be consistent with terminologies such as feed intake vs. feed consumption, to improve readability. One thing I noticed throughout the paper, is the use of “interesting strategy”, “interesting results” – I would avoid those statements and instead focus on the objective/actual metrics observed by the authors referenced.

-       While the authors did not see major effects of the treatments on the litter post-weaning performance, this should be included in the abstract, as it is stated in the title and is one of the main objectives of the study.

-       Finally, please check sentence structure throughout the discussion section, some sentences need some attention: “This feeding strategy has been widely adopted by commercial 358 pig farmers because of its simplicity and involves increasing feed intake at the end of ges-359 tation to meet the higher nutritional demands during this stage while still using one diet 360 for gestation [5].”

Specific comments

-       Line 10: Suggest “Sows are conventionally fed one diet throughout gestation formulated to meet the requirements of most sows. However, those diets can lead to nutritional excesses or deficiencies based on each sow's parity or stage of gestation.

-       Line 27-28: Would be possible to provide the amount of feed that was provided for FF and BF sows?

-       Line 32: Here and elsewhere – please use lower than or greater/higher than instead of “significantly lower/higher”. P-values are provided.

-       Line 36-37: Can you provide the p values for N and P intake? This is a major finding of the study so it should be moved to an earlier part of the abstract.

-       Line 53: delete globally

-       Line 55-56” please rephrase: “redistributing the energy intake during gestation without giving more overall 55 feed than a diet without bump feeding.

-       Line 56-61: Please rephrase this section to give it a better flow.

-       Line 64-65: Be specific: what interesting effects?

-       Line 65: Here and elsewhere, be consistent – feed consumption vs. feed intake.

-       Line 66-68: Be specific – what positive effects were observed by Quiniou et al.?  

-       Line 69: Here and elsewhere – please avoid statements such as “interesting results”, “interesting findings” and instead focus on the objective/actual metrics observed by the authors referenced.

-        

-       Line 96: Would be possible to provide average body weight with SD of sows and breed?

-       Line 99-105: The authors need to provide more details to the reader – diets are isoenergetic (how much energy?); constant feed quantity (how much?), lower feed intake (how much lower?). Also, again, please be consistent with terminologies (quantity, intake, etc.).

-       Line 109-110: “existing recommendations for the 109 herd’s genetics (Large White × Landrace crossbreed).” Reference?

-       Table 1: There are some values inside parenthesis in your table. Please indicate this in your footnotes.

-       Statistical analysis: Authors mentioned that they analyzed the data using ANOVA, but then mention fixed and random effects in the model. Could you please explain? Also, for post-weaning, I assume that the block was included as a random effect?

-       Table 2: I would suggest a multi-panel bar chart here – the Table as it is, is very confusing. Also, please check on the use of comma and dots.

-       Line 402: decrease on..

Round 2

Reviewer 1 Report

Comments and Suggestions for Authors

Thanks for your efforts on this revision.

Please review the cited reference numbers in Lines 58–75 of the manuscript to ensure they are in sequential order. For instance, references [10] and [11] have been cited before [9]. Check all the reference numbers throughout the manuscript.

Reviewer 3 Report

Comments and Suggestions for Authors

NA